# Carrying Temoporfin with Human Serum Albumin: A New Perspective for Photodynamic Application in Head and Neck Cancer

**DOI:** 10.3390/biom13010068

**Published:** 2022-12-29

**Authors:** Edoardo Jun Mattioli, Luca Ulfo, Alessia Marconi, Valentina Pellicioni, Paolo Emidio Costantini, Tainah Dorina Marforio, Matteo Di Giosia, Alberto Danielli, Carmela Fimognari, Eleonora Turrini, Matteo Calvaresi

**Affiliations:** 1Dipartimento di Chimica “Giacomo Ciamician”, Alma Mater Studiorum—Università di Bologna, Via Francesco Selmi 2, 40126 Bologna, Italy; 2Dipartimento di Farmacia e Biotecnologie, Alma Mater Studiorum—Università di Bologna, Via Francesco Selmi 3, 40126 Bologna, Italy; 3Dipartimento di Scienze per la Qualità della Vita, Alma Mater Studiorum—Università di Bologna, Corso d’Augusto 237, 47921 Rimini, Italy

**Keywords:** temoporfin (mTHPC), human serum albumin, head and neck squamous cell carcinoma cells, photodynamic therapy, reactive oxygen species (ROS), in vitro anticancer activity

## Abstract

Temoporfin (mTHPC) is approved in Europe for the photodynamic treatment of head and neck squamous cell carcinoma (HNSCC). Although it has a promising profile, its lipophilic character hampers the full exploitation of its potential due to high tendency of aggregation and a reduced ROS generation that compromise photodynamic therapy (PDT) efficacy. Moreover, for its clinical administration, mTHPC requires the presence of ethanol and propylene glycol as solvents, often causing adverse effects in the site of injection. In this paper we explored the efficiency of a new mTHPC formulation that uses human serum albumin (HSA) to disperse the photosensitizer in solution (mTHPC@HSA), investigating its anticancer potential in two HNSCC cell lines. Through a comprehensive characterization, we demonstrated that mTHPC@HSA is stable in physiological environment, does not aggregate, and is extremely efficient in PDT performance, due to its high singlet oxygen generation and the high dispersion as monomolecular form in HSA. This is supported by the computational identification of the specific binding pocket of mTHPC in HSA. Moreover, mTHPC@HSA-PDT induces cytotoxicity in both HNSCC cell lines, increasing intracellular ROS generation and the number of γ-H2AX foci, a cellular event involved in the global response to cellular stress. Taken together these results highlight the promising phototoxic profile of the complex, prompting further studies to assess its clinical potential.

## 1. Introduction

Head and neck squamous cell carcinomas (HNSCC) account for 90% of head and neck cancers [1]. They represent approximately 900,000 cases and over 400,000 deaths annually worldwide [2]. HNSCC arises from the squamous epithelium of the upper aerodigestive tract and mostly affects nasal cavity, pharynx, larynx, lips, and paranasal sinuses [1]. The risk factors most frequently associated with HNSCC include smoking, alcohol consumption, human papillomavirus (HPV) infection, and Epstein-Barr virus (EBV) infection. Patients affected by these carcinomas are usually diagnosed with advanced local disease, which often leads to recurrences and a poor prognosis [1].

Surgical resection represents the standard therapy for advanced HNSCC, followed by adjuvant radiation and/or platinum-based chemotherapy. These therapies are associated with severe adverse effects, such as xerostomia, oral mucositis, osteoradionecrosis, and dermal complications, which compromise patients’ quality of life. Moreover, despite the multimodal therapeutic strategy to fight HNSCC, patients have a poor response to therapy and treatment failure occurs in 40–50% of patients [3]. Thus, major efforts are focused on the identification of alternative therapeutic strategies to improve the treatment of HNSCC, especially for patients with recurrent or metastatic disease. In this context, photodynamic therapy (PDT) is gaining increasing interest [4].

PDT is a promising minimally invasive strategy that represents a valuable therapeutic procedure for the management of a variety of solid tumors. PDT uses photosensitizers (PSs) that, after being excited by light at a specific wavelength, interact with the molecular oxygen generating reactive oxygen species (ROS) in the target tissue, resulting in cell death [5]. Temoporfin (mTHPC) is a chlorin-based photosensitizer (Figure 1) that was approved in 2001 in Europe for the photodynamic treatment of HNSCC. 

mTHPC [6] is one of the most effective second-generation PS for clinical PDT and possesses good photophysical properties and high singlet oxygen yield. The first advantage of mTHPC is that it is applied at very low doses (0.1 mg kg^−1^) compared to other PSs and requires very low energy intensity (10 J cm^−2^) for clinical applications [6]. Despite the promising profile, some drawbacks restrain the use of mTHPC in clinical settings. First, its lipophilic character hampers the full exploitation of its potential, as its low water solubility and high tendency to aggregation cause very low stability in a physiological environment and reduce ROS generation, and thus, PDT efficacy. In addition, mTHPC is not selective for cancer cells, leading to unwanted phototoxicity [6].

mTHPC is administered intravenously to patients using a mixture of ethanol anhydrous and propylene glycol. Paravenous and intra-arterial injection of this formulation can cause severe erythema, oedema, and mild to moderate subcutaneous inflammation and hemorrhage [7]. For this reason, many studies aim to improve the solubility and stability of mTHPC in a physiological environment through new delivery systems based on polymers, liposomes, and nanoparticles [8]. 

Proteins are biological molecules that play many critical roles in the body. One of these roles is to bind and transport hydrophobic molecules inside the cells and throughout the body. Proteins can act as supramolecular hosts or drug delivery systems [9,10,11,12], conferring solubility to hydrophobic PS in physiological media [13,14,15,16,17,18,19,20,21,22]. In particular, human serum albumin (HSA) has been extensively explored as a versatile carrier for hydrophobic PSs via covalent conjugation [23,24,25] or supramolecular interactions [26,27,28,29].

HSA exhibits both i) passive tumor targeting due to its enhanced permeability and retention (EPR) effect, and ii) active targeting due to HSA receptors that are overexpressed on cancer cells, such as the albumin-binding protein SPARC (secreted protein acidic and rich in cysteine) and gp60 (a 60-kDa sialoglycoprotein) [30].

In the present paper we aim to explore the efficiency of a new mTHPC formulation that uses HSA to disperse the drug in solution and to investigate the mechanism that triggers inhibition of cancer cells viability in two HNSCC cell lines upon PDT treatment.

## 2. Materials and Methods

### 2.1. Synthesis and Characterization of the mTHPC@HSA Complex

#### 2.1.1. Materials

Human serum albumin fatty acid free (HSA) (Cat. No. A3782), 9,10-anthracenediylbis(methylene)dimalonic acid (ABMDMA) (Cat. No. 75068), deuterium oxide (Cat. No. 151882-100G), Amicon Ultra centrifugal filters (MWCO 30 kDa, Millipore UFC503024) (Cat. No. Z677892-24EA), dimethyl sulfoxide (DMSO) (Cat. No. 472301), ethanol (EtOH) (Cat. No. 34852), propylene glycol (PG) (Cat. No.Y0001547), ammonium acetate (NH_4_Ac) (Cat. No. A7262), sodium chloride (Cat. No. S9888-M), potassium phosphate monobasic (Cat. No. P0662-M), sodium phosphate dibasic (Cat. No. S0876), potassium chloride (Cat. No. P3911M), sodium bicarbonate (Cat. No. 31437-M), sodium carbonate (Cat. No. 223530), and MWCO 14 kDa dialysis tubing cellulose membrane (Cat. No. D9652) were purchased from Sigma Aldrich (Merck, Darmstadt, Germany). 3,3′,3″,3′′′-(7,8-dihydro-21*H*,23*H*-porphine-5,10,15,20-tetrayl)tetrakis-phenol (mTHPC) (Item No. 17333) was purchased from Cayman Chemical (Ann Arbor, MI, USA). All the reagents were used without further purifications. Milli-Q water was used for the preparation of all the aqueous solutions.

#### 2.1.2. Synthesis and Purification of mTHPC@HSA Complex

The complex mTHPC@HSA was synthesized by adapting the procedure previously described by Adams et al. [31]. mTHPC and HSA were used in 1:1 stoichiometry. 

Solution A. HSA was firstly dissolved in PBS, then DMSO was slowly added to the solution to obtain a final concentration of HSA 200 µM, dissolved in DMSO/PBS (3/5 *v*/*v*) mixture.

Solution B. A stock solution of mTHPC in DMSO was prepared. It was then diluted to a final concentration of 200 µM of mTHPC in a mixture of DMSO/PBS (3/5 *v*/*v*) just before the addition to the HSA solution.

A volume of 500 µL of Solution B was slowly added to Solution A under gentle stirring, obtaining a final solution where the concentration of both the components was 100 µM.

The mixture was then incubated overnight at 25 °C under continuous shaking at 700 rpm (ThermoMixer HC, S8012-0000; STARLAB, Hamburg, Germany). After incubation, the solution was extensively dialyzed against PBS, using a 14 kDa cutoff cellulose membrane dialysis tubes, to remove DMSO and possible free mTHPC. After the purification procedure, UV-Vis measurements showed a final stoichiometry of 0.75:1 mTHPC/HSA.

#### 2.1.3. Characterization of mTHPC@HSA Complex

UV-Vis Spectroscopy. mTHPC, HSA, and mTHPC@HSA were characterized through UV-Vis spectroscopy. The absorption spectra were recorded using a Cary60 UV-Vis spectrophotometer (Agilent Technologies, Stockport, UK). 

Fluorescence Spectroscopy. The fluorescence spectra were recorded with an Edinburgh FLS920 equipped with a photomultiplier Hamamatsu R928P.

Mass Spectrometry. Mass spectrometry characterizations were carried out in native conditions by direct flow injection using an ESI-QTOF mass spectrometer (Waters Corporation, Milford, Worcester, MA, USA). The samples buffer was previously exchanged by using Amicon Ultra centrifugal filters (UFC503024, MWCO 30 kDa, Millipore, Burlington, MA, USA), obtaining a final concentration of mTHPC@HSA complex of 25 μM in NH_4_Ac 10 mM buffer. Positive-ion ESI mass spectra were acquired by applying a capillary voltage of 3 kV and a sample cone voltage of 45 V. The desolvation gas flow was set at 800 L h^−1^. The source and desolvation temperatures were 150 and 350 °C, respectively. All spectra were acquired in the range of *m/z* 500–5000. Raw data were background-subtracted and deconvoluted using Unidec software 5.0.2 version (University of Oxford, Oxford, UK) in the range of *m*/*z* 1500–4500 and mass 5–140 kDa.

Agarose gel electrophoresis. mTHPC@HSA complex was characterized by agarose gel electrophoresis, in native conditions, using an Owl Easycast B-Series Horizontal Gel Systems Model B2. Agarose gel was prepared at 1% *w*/*v* concentration in tris-glycine buffer at pH 7.4. A ratio of 20% *v*/*v* of glycerol was added to the samples dissolved in PBS, and 12 µL of the mixture were loaded into each well. The protein amount loaded was 15 µg, and mTHPC concentration in its reference samples was equimolar (20 µM) to the amount of mTHPC contained in the complex. Tris-glycine at pH 7.4 was used as running buffer, and the run was performed by applying a voltage of 100 V for 30 min.

Dynamic light scattering. All the measurements were performed by using a Malvern Instruments DLS ZetaSizer Nano-ZS (Malvern Panalytical Ltd., Malvern, Worcestershire, UK). mTHPC@HSA complex and mTHPC dissolved in PBS from the clinical formulation (EtOH/PG (40/60, *w*/*w*)) or from the DMSO stock solution were used at the same concentration used for UV-Vis characterization.

#### 2.1.4. Detection of Singlet Oxygen Generation

A colorimetric assay was carried out to selectively detect the amount of singlet oxygen (^1^O_2_) produced. The assay used 9,10-anthracenediyl-bis(methylene) dimalonic acid (ABMDMA) as a molecular probe. The reaction of ABMDMA with ^1^O_2_ produces its corresponding endoperoxide and can be monitored by UV-Vis spectrophotometry. The characteristic UV absorption band of ABMDMA progressively decreases as the corresponding endoperoxide (not absorbing species) is formed, allowing the estimation of the singlet oxygen produced. 

The sample buffer was exchanged with PBS dissolved in D_2_O, using Amicon Ultra centrifugal filters (UFC503024, MWCO 30 kDa, Millipore, Burlington, MA, USA). A 5 mM stock solution of ABMDMA was prepared in DMSO. A volume of 97 µL of the samples was loaded to a 96 multiwell plate, then 3 µL of ABMDMA stock were added to the sample. The multiwell plate was exposed to the light source (Valex 30W, 6500 K, cold white LED, see Appendix A for the spectral profile of the light source), at 19 cm distance from the cell-plate (irradiance = 24 mW cm^−2^, energy fluence = 86 J cm^−2^, measured with the photo-radiometer Delta Ohm LP 471 RAD). 

Based on the relative decrease of the absorbance of the ABMDMA (150 µM) before and after irradiation, recorded at 380 nm, we estimated the amount of the produced singlet oxygen.

### 2.2. Computational Investigation of the mTHPC@HSA Complex

#### 2.2.1. Ensemble Docking

All the available crystal structures of HSA were downloaded from the Protein Data Bank (PDB) (https://www.rcsb.org/). The PDB files were processed by removing water molecules, ions, and co-crystallized ligands. This dataset was used for ensemble docking calculations. All the possible orientations of the mTHPC phenol groups with respect to the chlorin plane were considered in the docking studies. Ensemble docking calculations were carried out considering all these conformers and every HSA PDB structure.

Docking models were obtained using the PatchDock algorithm Beta 1.3 Version [32]. 948,220 docking poses were generated and sorted using the PatchDock scoring function. The best pose was selected as input for MD simulations.

#### 2.2.2. MD Simulations

All MD simulations were carried out using the Amber16 software package [33]. The Amber ff14SB force field was used to model HSA. mTHPC atoms were modelled using the GAFF force field, and atomic charges were determined using the Merz–Singh–Kollman scheme. The corresponding parameters were generated by the standard procedure reported for antechamber, as implemented in Amber16 [34]. All simulations and minimization were performed using the TIP3P water model, and sodium counterions were added to maintain the electric neutrality of the system. Periodic boundary conditions (PBC) and the particle mesh Ewald summation were used throughout (with a cut-off radius of 10 Å). H-atoms were considered using the SHAKE algorithm and a time step of 2 fs was set during all the MD runs. A total of 500 steps of steepest descent minimization, followed by an additional 9500 steps of conjugate gradient minimization, were performed with PMEMD [34]. The minimized structure was subject to an equilibration process of 10 ns, heating the system to 298 K using an NPT ensemble and temperature coupling according to Andersen. Subsequently, 1 μs of MD trajectory was produced. Snapshot structures were saved into individual trajectory files every 1000 time steps, i.e., every 2 ps of the MD simulation. Secondary structure timeline analysis was computed by the timeline plugin contained in VMD [35].

#### 2.2.3. Molecular Mechanics/Generalized Born Surface Area (MM/GBSA) Analysis

A total of 10,000 frames were extracted from the MD trajectory by means of CPPTRAJ [34] and used as input for the MM/GBSA analysis to calculate the binding affinity between mTHPC and HSA. An infinite cut-off was used for all the interactions. The electrostatic contribution to the solvation free energy was calculated using the generalized Born (GB) model, as implemented in MMPBSA.py [36]. The non-polar contribution to the solvation free energy was determined using solvent-accessible surface-area-dependent terms.

### 2.3. Cytotoxicity and Phototoxicity of mTHPC@HSA Complex in HNSCC Cells

#### 2.3.1. Cell Lines

HNSCC cells lines CAL27, derived from squamous carcinoma of the oral tongue, and SQD9, from laryngeal squamous carcinoma, were kindly gifted by the laboratory of Experimental Radiotherapy, Leuven, Belgium. Cells were propagated in culture in DMEM high glucose cell medium, supplemented with 10% heat-inactivated fetal bovine serum, 1% L-glutamine solution 200 mM, and 1% penicillin/streptomycin solution 100 U mL^−1^ (all provided by Euroclone, Pero, Italy). Cells were cultured at 37 °C in a 5% CO_2_ humidified incubator. To maintain exponential growth, cells were trypsinized before reaching 80% confluence using the cells dissociation solution Versene (Merck, Sigma-Aldrich, Darmstadt, Germany). 

#### 2.3.2. Cell Viability

Cells were treated in complete medium with increasing concentrations of mTHPC/PBS (from the DMSO stock solution) or mTHPC@HSA (0.01–1.00 µM) for 45 min. At the end of incubation time, cells were washed twice in PBS 1X and irradiated in PBS 1X with a low irradiance white light LED (24 mW cm^−2^) for 45 min. 

In parallel, to check possible dark toxicity, cells were exposed to mTHPC/PBS or mTHPC@HSA but kept in the dark. After irradiation or dark incubation, cells were recovered for 24 h in complete medium and then cell viability was analyzed using the colorimetric MTT (3-(4,5-dimethylthiazol-2-yl)-2,5-diphenyltetrazolium bromide) assay (Merck). Briefly, cells were exposed to 0.5 mg mL^−1^ MTT solution in fresh medium for 90 min in the incubator. At the end of incubation time, MTT solution was removed and DMSO was added for the solubilization of formazan salts, whose formation is proportional to cell viability. Absorbance was measured at 570 nm using Victor X3 multimodal plate reader (Perkin Elmer, Walthman, MA, USA). The IC_50_, meaning the inhibitory concentration causing 50% of cell toxicity, was calculated for CAL27 and SQD9 cells after 24 h from treatment, by interpolation from a dose response curve.

#### 2.3.3. Analysis of Intracellular Uptake

The intracellular uptake of mTHPC@HSA and mTHPC/PBS was determined by flow cytometry. Briefly, CAL27 and SQD9 cells were incubated with mTHPC@HSA and mTHPC/PBS at two concentrations (0.3 and 1 µM) for 45 min in fresh complete medium, while controls were incubated just with fresh medium. At the end of incubation, 20,000 cells were used for the analysis of each experimental condition. Samples were washed twice after incubation and fixed with 4% paraformaldehyde for 15 min, then washed once and stored in 1X PBS at room temperature. mTHPC-related fluorescence was measured in the APC channel (660/20 nm). The analyses were performed on a Beckman Coulter CytoFLEX S (Brea, CA, USA). At least 2000 events were recorded for each analyzed sample. Data analysis was processed with FlowJo™ 10.0.7r2 version (Becton Dickinson, Franklin Lakes, NJ, USA).

#### 2.3.4. Analysis of Cell Death Mechanisms

To investigate the mechanism of cell death evoked by mTHPC@HSA, after 24 h from treatment, cells were washed, gently trypsinized using Versene, and incubated for 20 min with 100 µL of Guava Nexin Reagent (Luminex, Austin, TX, USA) at room temperature in the dark. The Guava Nexin assay allows the assessment of an early event in programmed cell death that is the translocation of phosphatidylserine from the inner to the outer side of the cytoplasmic membrane. The analysis involves the use of annexin-V, conjugated to phycoerythrin, which specifically binds to phosphatidylserine. However, even in cells that die by non-programmed death, such as necrosis, or in the later stages of programmed cell death that are characterized by damaged cell membrane, annexin-V binds to phosphatidylserine present on the inner side of the cytoplasmic membrane. This necessitates the use of the second dye, 7-aminoactinomycin (7-AAD), which intercalates with the DNA but can only penetrate cells with a damaged membrane. After staining with annexin V/phycoerythrin and 7-AAD, three cell populations can be distinguished: living cells (annexin V/phycoerythrin-/7-AAD-), cells undergoing programmed cell death in the early stages (annexin V/phycoerythrin+/7-AAD-), and necrotic cells or cells in the later stages of programmed cell death (annexin V/phycoerythrin+/7-AAD+). Cells were analyzed using the flow cytometer Guava EasyCyte 6-2L (Luminex). 

#### 2.3.5. Intracellular ROS Generation

To assess intracellular ROS generation, 20,000 CAL27 or SQD9 cells were seeded in a 96 well-plate. Cells were treated and incubated in complete medium with increasing concentration of mTHPC@HSA for 45 min. After incubation, cells were washed twice with PBS 1X, to remove any excess of the drug. Cells were irradiated for 45 min in PBS 1X. In parallel, to check eventual ROS generation in dark condition, cells were treated according to the aforementioned conditions and kept in the dark. After irradiation, 100 µL of ROS-Glo (Promega, Madison, WI, USA) were added to each well and cells were incubated for 20 min at room temperature. Luminescence was recorded using EnSpire^®^ Multimode plate reader (Perkin Elmer). Results are expressed as fold increase compared to untreated cells.

#### 2.3.6. H2AX Phosphorylation Assay

To investigate whether mTHPC@HSA was able to induce cellular stress in HNSCC cells, the phosphorylation of H2AX (γ-H2AX) was analyzed. Briefly, after mTHPC@HSA treatment (45 min treatment/45 min irradiation) and 24 h of recovery in complete medium, HNSCC cells were fixed with paraformaldehyde 4% and permeabilized using cold methanol 90%. After permeabilization, cells were washed with incubation buffer (PBS 1X added with 1% BSA) and incubated for 30 min with the anti-γ-H2AX-FITC conjugate antibody (Merck). Samples were analyzed via flow cytometry and results were expressed as fold increase of γ-H2AX in treated cells versus untreated cells. 

#### 2.3.7. Statistical Analysis

Results are expressed as the mean ± SEM of at least three independent experiments. The analysis of variance with Dunnett’s or Kruskal–Wallis as posttest or *t* test were used. The statistical software GraphPad InStat 8.0 version (GraphPad Prism, San Diego, CA, USA) was used. In this work, *p* < 0.05 was considered significant.

## 3. Results and Discussion

### 3.1. Synthesis and Characterization of THPC@HSA Complex

HSA is able to bind and disperse mTHPC [37,38]. Their interaction has been characterized by a variety of techniques including fluorescence, UV-Vis absorption, Fourier transform infrared, circular dichroism, and EPR [39,40,41]. Here we synthesized a mTHPC@HSA complex, with a well-defined 1:1 stoichiometry, using a PBS/DMSO mixed solvent system, with overnight incubation and purification by dialysis (See Materials and Methods section for details). The UV-Vis spectrum of mTHPC@HSA (Figure 1) demonstrated the success in the incorporation of mTHPC in HSA. In fact, it revealed features that belong to both components of the complex, showing the distinctive absorption bands of mTHPC (i.e., the Soret band at 426 nm and the four Q bands in the range of 500–680 nm) and of the protein (281 nm).

The electrophoretic characterization of mTHPC@HSA, performed on agarose gel in non-denaturing conditions, clearly confirms the encapsulation of mTHPC in the protein (Appendix A). In addition, the mass spectrum of the mTHPC@HSA complex, yielding a peak corresponding to the mass of HSA plus the mass of mTHPC (Appendix A), proved the successful supramolecular dispersion of mTHPC with the protein.

mTHPC in its pure clinical formulation (EtOH/PG (40/60, *w*/*w*)) can be considered as a reference for monomolecular dispersion of mTHPC. The mTHPC@HSA complex in PBS shows the characteristic UV-Vis spectrum of mTHPC monomers, maintaining a well-defined shape of the Soret band centered at 426 nm and the four absorption Q-bands in the 500–680 nm range [42]. Conversely, the dispersion of the same quantity of mTHPC in PBS, starting from the clinical formulation of the mTHPC (EtOH/PG (40/60, *w*/*w*)) or from a stock solution in DMSO, suggested that mTHPC exists mainly in aggregated form, as evident from the broad and slightly red-shifted Soret band (at 440 nm) (Figure 2a). Importantly, the aggregation state of mTHPC has a strong influence on the photosensitizing and imaging properties of mTHPC. The aggregated state significantly affects both ROS generation and fluorescence emission of mTHPC by quenching [37,43,44]. On the contrary, mTHPC monomers are not affected by quenching and present a much stronger fluorescence emission [43,44]. In agreement with the UV-Vis spectrum, when isoabsorbing solutions of mTHPC were dispersed monomolecularly by HSA, the characteristic emission of mTHPC at 656 nm was observed (inset in Figure 2a); otherwise, when mTHPC was in aggregated state, no fluorescence could be detected.

DLS analysis of the solutions (Figure 2b) confirmed that mTHPC@HSA in PBS is monomeric and characterized by a size of ~8 nm, corresponding to the hydrodynamic diameter of the HSA protein, encapsulating a single mTHPC molecule (Appendix A). At the same concentration, the mTHPC obtained from the clinical formulation and from the DMSO stock solution leads to aggregates with a medium size of 530 nm and 1280 nm, respectively (Figure 2b).

Interestingly, dilution of the clinical formulation of mTHPC in PBS leads to three different dispersion phases: the pure clinical formulation shows monomeric dispersion of mTHPC until 1:2 dilution; monomers and aggregates coexist at the 1:4 dilution; then, increasing the PBS content, mTHPC is present only in the form of aggregates (Appendix A). During clinical use, the dilution of the mTHPC clinical formulation is well beyond the 1:4 value, suggesting the formation of mTHPC aggregates. 

The mTHPC@HSA complex proved stable for long periods when stored in the dark (at least up to two weeks, Appendix A) and upon dilution (Appendix A). The stability of mTHPC@HSA was also measured in a medium containing 10% FBS. Equilibrium dialysis experiments (Appendix A) demonstrated that mTHPC@HSA does not exchange the mTHPC molecule with the free serum proteins, as opposed to the clinical formulation mTHPC which rapidly binds to the serum proteins.

### 3.2. Atomistic Details of the mTHPC@HSA Complex

To have an atomistic detail of the interaction between mTHPC and HSA, an ensemble docking calculation, followed by MD simulations, were carried out. The structure of HSA consists of three domains (D_I_, D_II_, and D_III_, see Appendix A). Seven binding pockets for fatty acid (FA1–7) were identified. Drugs usually bind HSA at two major binding sites, Sudlow’s site I (FA7) and site II (FA3, FA4). The largest FA pocket is the heme binding site (FA1). The cleft region between D_I_ and D_III_ forms an additional cavity where large molecules can bind (Appendix A).

The docking calculations suggested that the large mTHPC molecule binds in the cleft region (Figure 3A), as already observed for phthalocyanine derivatives [45,46].

The heme binding pocket is not ample enough to accommodate the mTHPC molecule, due to the presence of the four additional benzene rings in the molecular structure of the mTHPC. A larger cavity in the protein, such as the one existing in the cleft region, is necessary to bind mTHPC.

Starting from this assumption, we carried out a 1 μs MD simulation, followed by MM/GBSA analysis of the trajectory to investigate the binding between mTHPC and HSA. The binding process between mTHPC and HSA is a favored process with a ΔE_binding_ = −36.4 kcal mol^−1^. mTHPC showed a strong shape complementarity with the HSA pocket in the cleft (Figure 3B). A direct consequence is that van der Waals (vdW) interactions (E_vdW_ = −54.4 kcal mol^−1^) are the most important term for the recognition and binding of mTHPC to HSA. The formation of hydrogen bonds between mTHPC and HSA is suggested by stabilizing electrostatic interactions (E_Elec_ = −35.0 kcal mol^−1^).

We also analyzed the effect of mTHPC binding in terms of solvation contributions. The binding of a hydrophobic molecule, such as mTHPC, in a protein cavity explains the non-polar solvation term (E_non-polar solvation_ = −7.3 kcal mol^−1^) assisting the binding (hydrophobic effect). On the contrary, the binding of mTHPC occurs in a region that is normally exposed to water and where amino acids with polar side groups are located. These residues, in the presence of the hydrophobic mTHPC, are forcedly desolvated and are no longer able to interact with water molecules usually present in this region, causing a destabilization of the system, so the polar solvation term is detrimental for the binding and its contribution is positive (E_polar solvation_ = 60.3 kcal mol^−1^).

The results of the simulations agree well with EPR experiments [41] that determined: (i) a distance between the mTHPC binding site and the Cys-34 residue of HSA of ~2.9 ± 0.25 nm, and (ii) the location of mTHPC deep inside the HSA and almost completely shielded from the solvent molecules. 

The decomposition analysis (fingerprint analysis) of the overall binding energy provides also the contribution of each amino acid of the HSA to the binding of mTHPC. Analyzing the most important interactions (Figure 4), Asp108 and Asp187 interact with mTHPC, forming a hydrogen bond between the carboxylic group of the side chain of the residues and the hydroxyl groups of the phenol moiety in mTHPC. The interactions between Ala194, Gln 459, and Val462 and mTHPC are instead mainly VdW.

The effect of mTHPC loading on HSA structure was determined by analysis of the secondary structure of the protein in MD simulations. While the electrophoretic characterization of mTHPC@HSA confirmed the absence of aggregation phenomena (Appendix A) and DLS experiments determined that the tridimensional structure of the HSA protein was not perturbed by the presence of the mTHPC (Appendix A), the analysis of the MD trajectory allowed an atomistic view of the interaction between mTHPC and HSA. The comparison of the secondary structure of HSA and mTHPC@HSA during MD simulations (Appendix A) showed that the 3D structure of HSA during the MD simulation is practically unaffected by the mTHPC binding. In fact, 17 disulfide bridges impart rigidity to the helical, globular structure of HSA. Locally a small variation is observed upon mTHPC binding on the terminal C-terminal helices in domain III. In the crystal structure of HSA, these helices are characterized by very high temperature factors [47], and their mobility is commonly observed during complex formation [47]. The insertion of mTHPC into the hydrophobic cleft of HSA, mostly through hydrophobic interactions, may be responsible for the small reduction of α-helicity of HSA, as also observed experimentally with FT-IR and CD spectroscopy [39].

These results indicate that HSA, much alike a “Trojan Horse” [48], efficiently encapsulates a mTHPC molecule into a specific pocket, dispersing the hydrophobic molecule in a monomolecular form in physiological environment, and giving a well-defined biological identity to the PS.

### 3.3. mTHPC@HSA Generates Singlet Oxygen in PBS

The ability of the mTHPC@HSA complex to behave as PS, upon irradiation with visible light in the physiological environment, was evaluated using ABMDMA as a probe to detect ^1^O_2_ production [49]. ABMDMA reacts with ^1^O_2_ to give an endoperoxide. This reaction can be monitored by the disappearance of the ABMDMA absorption band. We measured the amount of ^1^O_2_ generated upon visible light irradiation in PBS by mTHPC@HSA, comparing the results with the solutions of mTHPC, prepared both from the clinical formulation of mTHPC and from the mTHPC stock solution in DMSO, all at the same concentrations.

When the mTHPC solutions prepared from the clinical formulation of the mTHPC or from the stock solution in DMSO were irradiated, only a small generation of ^1^O_2_ was detected (Figure 5). In contrast, a large quantity of ^1^O_2_ is generated in the PBS solution during the irradiation of the mTHPC@HSA (Figure 5). 

PSs are inactivated by aggregation. The triplet excited state, which produces singlet oxygen upon interaction with the ground state of molecular oxygen, is extremely sensitive to aggregation phenomena. Aggregation significantly quenches or decreases the lifetime of the long-lived triplet excited state, reducing the production efficiency of ^1^O_2_. In general, when proteins are used as supramolecular hosts for the dispersion of PS, the generation of ROS increases [14,15,16,17,18]. Here, the formation of the mTHPC@HSA complex clearly shows an enhancement in the ^1^O_2_ production yield due to: (i) monomolecular dispersion of the PS, and (ii) confinement of PS in the HSA binding pocket; the hydrophobic environment shields the sensitizing molecules from quenching by water molecules [50,51].

The developed formulation of mTHPC (mTHPC@HSA) is safe, biocompatible, stable in physiological environment, does not aggregate, and is extremely more performant in PDT that the standard clinical formulation. 

### 3.4. mTHPC@HSA Undergoes Intracellular Uptake and Decreases HNSCC Cells Viability upon PDT Treatment

We tested the cytotoxicity and the potential phototoxicity of the mTHPC@HSA complex in vitro, comparing the performances with mTHPC, to assess the efficacy of the monodispersed formulation versus the presence of aggregates.

We demonstrated that mTHPC/PBS did not significantly decrease cell viability either in dark conditions nor after irradiation, except for the highest tested concentration of PS (1 µM) in CAL27 cells, with 54% of viable cells (compared to 100% of untreated cells) (Figure 6). However, treatment of HNSCC cells with increasing concentrations of mTHPC@HSA strongly affected cell viability after irradiation in both cell lines, in a statistically significant manner compared to untreated cells and to mTHPC/PBS starting from the concentration 0.3 µM (Figure 6). In particular, the recorded cell viability was 21.4% for CAL27 and 31.1% for SQD9 after treatment with 0.3 µM mTHPC@HSA. Dark toxicity was negligible for all tested concentrations in both cell lines. The calculated IC_50_ after 24 h from 45 min incubation and 45 min irradiation with the white light LED (24 mW cm^−2^) was 0.18 µM for CAL27 and 0.23 µM for SQD9 cells.

The different intracellular uptake of the two complexes provides an explanation for the different cytotoxic activity observed for mTHPC@HSA and mTHPC/PBS (Figure 7). Indeed, cellular uptake was recorded only for cells incubated with mTHPC@HSA, where a concentration-dependent increase in fluorescence was observed for both HNSCC cell lines, compared to untreated cells. On the contrary, only a marginal increase in fluorescence was shown for samples incubated with mTHPC/PBS. These results confirmed that mTHPC@HSA exploits the ability of HSA to target tumor cells through its receptor-mediated uptake [30]. 

Cells were then analyzed through an annexin-V assay to have a preliminary insight into the mechanism of cell death triggered by mTHPC@HSA. Strongly depending on the concentration of PS, we recorded a concentration-dependent increase in annexin-V and 7-AAD positive cells (Figure 8). Cells treated with the highest concentration of mTHPC@HSA treatment were almost all double positive cells, meaning cells in later stages of programmed cell death or more probably dying for unprogrammed, necrotic, cell death (Figure 8).

In vitro results obtained on mTHPC showed that the switch from regulated, such as apoptosis and autophagy, and unprogrammed cell death is strictly dependent on light dose, PS concentration, and the cancer cell type [52]. It is well known that PDT therapy mediated by mTHPC induces oxidative stress that provokes tumor site necrosis and slough in HNSCC patients [7,53]. Therefore, it is not surprising to record necrotic events induced by irradiated mTHPC@HSA in HNSCC cell lines. 

Recalling that mTHPC is not able to exploit its cytotoxic activity in PBS solution, mainly due to its hydrophobic nature [6], our results indicate that mTHPC exerts its anticancer potential only when HSA is used for drug dispersion and delivery in tested HNSCC tumor cells. 

### 3.5. mTHPC@HSA Increases Intracellular ROS Generation upon PDT Treatment

To elucidate the mechanism responsible for mTHPC@HSA-induced cell death after irradiation, the intracellular ROS generation was assessed in HNSCC cell lines. At the highest tested concentration, we recorded a three-fold increase of intracellular ROS generation in CAL27 cells, whereas only a trend of increase was recorded for SQD9 cells (Figure 9). 

The different ability of the same PS, and in particular of mTHPC, to differently generate ROS in different cell lines, with the same treatment conditions, was already observed [52]. Indeed, different concentrations and different light doses were required to increase the generation of intracellular ROS among the cell lines [52]. In particular, the two HNSCC cell lines BHY and KYSE-70, cells of oral and esophagus cancer respectively, showed different ability to generate peroxides; BHY required higher light dose to generate ROS compared to KYSE-70, both tested at their IC_90_ concentration of mTHPC, meaning the inhibitory concentration causing 90% of cell toxicity [52]. One explanation on the differences recorded in ROS generation may be the different efficiency of the antioxidant pathways between cancer cells, for instance different levels of catalase, glutathione (GSH), or GSH-recovery enzymes [52,54]. Furthermore, it was previously demonstrated that inhibition of antioxidant pathways in HNSCC cells, such as GSH and thioredoxin, influences the response to anticancer therapy involving oxidative stress, but to a different extent, given the intrinsic differences of HNSCC cell types. For instance, CAL27 cells, tongue squamous cell carcinoma, possess a significantly higher level of catalase activity compared to FaDu cells, a hypopharyngeal carcinoma cell line that influences the cell killing rescue by treatment with buthionine sulfoximine and auranofin, inhibitors of GSH and thioredoxin, respectively [55]. Although only a small increase in ROS generation was observed in SQD9 cells, laryngeal squamous cell carcinoma, we recorded a statistically significant PDT-induced cell death, comparable to that recorded in CAL27 cells (Figure 6). 

### 3.6. mTHPC@HSA Induces γ-H2AX upon PDT Treatment

To further investigate the mechanisms underpinning the decrease in viability induced by mTHPC@HSA, its ability to induce cellular stress was investigated through the analysis of H2AX phosphorylation (Figure 10). H2AX is one of the key components of chromatin involved in DNA damage response. Its phosphorylation on serine 139 is one of the earliest events that rapidly concentrated in chromatin domain around DNA double strand breaks induced by cellular stress [56]. 

In line with previously published data on mTHPC [57], mTHPC@HSA per se did not interfere with γ-H2AX phosphorylation, but only after irradiation (data not shown). Conversely, mTHPC@HSA-PDT significantly increased the number of γ-H2AX foci in both HNSCC cell lines (Figure 10). The increase in fluorescence recorded after irradiation proved to be proportional to the PS concentrations. This suggests a positive correlation with the PS amount and the PDT-induced oxidative stress, as indicated by the impairment of DNA repair systems [58,59]. The increase in DNA repair systems we observed after 24 h from photoactivation agrees with the results of two previous studies. A first study showed DNA single strand breaks immediately after treatment with photoactivated mTHPC that were repaired after 4 h from treatment in K562 leukemia cells [60]. A second study demonstrated damage to DNA immediately and after 4 h from photoactivation, which was repaired after 24 h, as indicated by the increase in DNA repair system [58]. Furthermore, we interestingly observed these results in CAL27 and SQD9, which are HPV-negative tumors. However, HPV-positive patients, more than HPV-negative, are usually characterized by defects in the signaling and repair of DNA double-strand breaks and this observation correlates with increased responsiveness to chemotherapy and radiotherapy [61]. In light of this consideration, we hypothesize that mTHPC@HSA may exhibit an even better efficacy in HPV-positive cells.

## 4. Conclusions and Perspectives

The dispersion of mTHPC in HSA exploits the versatility of this protein as “Trojan Horse” for drug delivery able to preferentially accumulate in tumor tissue [30]. Together with the passive targeting mediated by the EPR effect, the accumulation of HSA in tumor tissue may be mediated by specific albumin binding proteins that accumulate in tumors, such as SPARC or gp60 [62]. SPARC is a multifunctional glycoprotein associated with tumor development, invasion, metastasis, and prognosis. It is highly expressed in several malignant tumors, including HNSCC [63] and is a poor prognostic factor for this type of tumor. The cellular uptake of HSA is enhanced by SPARC-mediated endocytosis in oral squamous cell cancer (OSCC) cell line [64]. Previous studies demonstrated that the treatment of HNSCC patients with Abraxane (nab-paclitaxel), the first approved chemotherapeutic nanotechnological formulation based on HSA, may correlate with SPARC overexpression, converting the SPARC-positive patients in that one showing the better clinical outcome [63,65]. This evidence points out the usefulness of HSA formulation for chemotherapeutic drugs and mTHPC@HSA perfectly fits in this context. Moreover, the performances of the mTHPC@HSA platform can be additionally improved by using light-harvesting antennae to extend the PDT activity of the mTHPC into the NIR [66], or by attaching targeting moieties to specifically address cellular receptors that are usually overexpressed in cancer cells for receptor-targeted PDT [67,68,69,70,71].

In this study we demonstrated, through a comprehensive characterization, that mTHPC@HSA is stable in a physiological environment, does not aggregate, and is extremely efficient in PDT performance due to its high singlet oxygen generation and the high dispersion as monomolecular form in HSA. This is supported by the computational identification of the specific binding pocket of mTHPC in HSA. Moreover, mTHPC@HSA-PDT induced cytotoxicity in both tested HNSCC cell lines. We ascribe the recorded cell death to the ability of the complex to increase intracellular ROS generation, although in a different extent in the two analyzed cell lines, and to increase the phosphorylation of H2AX related to oxidative stress.

Taken together, these data highlight the promising phototoxic profile of mTHPC@HSA, prompting further studies to assess its clinical potential. The next steps for the translation from bench side to bedside need the completion of preclinical studies to determine the pharmacokinetics, the toxicological profile, and the PDT performances of the new formulation using in vivo models. Indeed, although our in vitro and in silico studies depict how promising mTHPC@HSA is, further studies are required for its future development.

## Data Availability

All data in this study can be requested from the corresponding authors (eleonora.turrini@unibo.it, E.T. and matteo.calvaresi3@unibo.it, M.C.).

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
