# Peer review of "Carrying Temoporfin with Human Serum Albumin: A New Perspective for Photodynamic Application in Head and Neck Cancer"

_biomolecules, 2022, doi:10.3390/biom13010068_

Round 1

Reviewer 1 Report

The article contains new interesting and unusual data concerning the possibility of using albumin as a carrier for the photosensitizer temoporphin. Until now, it was believed that this photosensitizer in the blood binds almost exclusively to lipoproteins and does not form complexes with serum albumin. The authors of the article used a method for loading albumin with a photosensitizer, which is usually used to produce polymer nanoparticles loaded with drugs. Using a number of experimental and computational methods, they concluded that monomeric photosensitizer molecules were embedded in a protein globule and evaluated their photodynamic activity. Regarding this part of the article, a number of questions arise, the answer to which is necessary for judging the conclusions made.

1. A quantitative assessment of the degree of protein loading with a photosensitizer after the purification procedure should be given.

2. Absorption spectra often do not provide complete information about the aggregation state of tetrapyrrole compounds. The use of the fluorescent characteristics of the photosensitizer makes it possible to more accurately characterize both its aggregation state and the features of the binding processes in the protein globule.

3. It is desirable to provide data concerning the effect of the loading procedure on the protein structure.

4. It can be assumed that the interaction of photosensitizer and protein is an example of equilibrium binding. In this regard, it is very important to evaluate the binding constants, as well as the dissociation rate constants of photosensitizer molecules from complexes. Knowledge of these parameters will make it possible to predict the behavior of albumin and photosensitizer complexes after injection into a medium containing serum.

5. From my point of view, it is of interest to study changes in the properties of solutions of complexes during their storage and dilution.

6. The use of chemical trap techniques to compare the efficiency of singlet oxygen generation in solutions and in nanostructured materials involves taking into account the distribution of trap molecules in the sample under study. The significant differences in the rate of singlet oxygen generation with the participation of mTHPC and mTHPC-protein established in the article may reflect not so much differences in sensitizing properties as the uneven distribution of trap molecules in the protein solution.

The second part of the article is devoted to the study of the processes of sensitization by mTHPC-albumin complexes of cell damage in culture. This part also contains interesting and unusual results. The credibility of these results, as well as the conclusions drawn from them, would be higher if the authors provided information on the following issues.

1. How can mTHPC-albumin complexes increase the delivery of a sensitizer to cells in the presence of a large amount of native albumin in the medium?

2. From my point of view, in order to understand the features of the interaction of the studied form of the sensitizer with various types of cells, as well as the mechanisms of their photosensitization, it is extremely important to study the quantitative characteristics of the accumulation of mTHPC in cells (for example, by flow cytometry).

3. It is believed that photosensitization of cells by mTHPC is not accompanied by significant damage to the genetic structures of cells. In this regard, what is the reason for the high yield of DNA damage when using albumin and photosensitizer complexes.

Author Response

Reviewer: 1

The article contains new interesting and unusual data concerning the possibility of using albumin as a carrier for the photosensitizer temoporphin. Until now, it was believed that this photosensitizer in the blood binds almost exclusively to lipoproteins and does not form complexes with serum albumin. The authors of the article used a method for loading albumin with a photosensitizer, which is usually used to produce polymer nanoparticles loaded with drugs. Using a number of experimental and computational methods, they concluded that monomeric photosensitizer molecules were embedded in a protein globule and evaluated their photodynamic activity. Regarding this part of the article, a number of questions arise, the answer to which is necessary for judging the conclusions made.

Comments:

  1.  A quantitative assessment of the degree of protein loading with a photosensitizer after the purification procedure should be given.

Authors’ reply:

We added the requested information in the manuscript.

We added a sentence that reads:

“After the purification procedure UV-Vis measurements showed a final stoichiometry of 0.75:1 mTHPC/HSA”

Comments:

  1. Absorption spectra often do not provide complete information about the aggregation state of tetrapyrrole compounds. The use of the fluorescent characteristics of the photosensitizer makes it possible to more accurately characterize both its aggregation state and the features of the binding processes in the protein globule.

Authors’ reply:

We followed the suggestion of the reviewer and we added emission spectra of the different mTHPC formulations (Figure 2A, inset).

We added in the manuscript a sentence that reads:

“Importantly, the aggregation state of mTHPC has a strong influence on the photosensitizing and imaging properties of mTHPC. The aggregated state significantly affects both ROS generation and fluorescence emission of mTHPC by quenching [39,45,46]. On the contrary, mTHPC monomers are not affected by quenching and present a much stronger fluorescence emission, [45,46]. In agreement with the UV-Vis spectrum, when isoabsorbing solutions of mTHPC were dispersed monomolecularly by HSA, the characteristic emission of mTHPC at 656 nm was observed (inset in figure 2a), otherwise, when mTHPC was in aggregated state no fluorescence could be  detected.”

Comments:

  1. It is desirable to provide data concerning the effect of the loading procedure on the protein structure.

Authors’ reply:

We provided additional data and discussion about the effect of the loading procedure on the protein structure.

We added a sentence in the manuscript that reads:

“The effect of mTHPC loading on HSA structure was determined by analysis of the secondary structure of the protein in MD simulations. While the electrophoretic characterization of mTHPC@HSA confirmed the absence of aggregation phenomena (Figure S2) and DLS experiments determined that the tridimensional structure of the HSA protein was not perturbed by the presence of the mTHPC (Figure S4), the analysis of the MD trajectory allowed an atomistic view of the interaction between mTHPC and HSA. The comparison of the secondary structure of HSA and mTHPC@HSA during MD simulations (Figure S10) showed that the 3D structure of HSA during the MD simulation is practically unaffected by the mTHPC binding.  In fact, 17 disulfide bridges impart rigidity to the helical, globular structure of HSA. Locally a small variation is observed upon mTHPC binding on the terminal C-terminal helices in domain III. In the crystal structure of HSA, these helices are characterized by very high temperature factors [50], and their mobility is commonly observed during complex formation [50]. The insertion of mTHPC into the hydrophobic cleft of HSA, mostly through hydrophobic interactions, may be responsible for the small reduction of -helicity of HSA, as also observed experimentally with FT-IR and CD spectroscopy [41].”

Comments:

  1. It can be assumed that the interaction of photosensitizer and protein is an example of equilibrium binding. In this regard, it is very important to evaluate the binding constants, as well as the dissociation rate constants of photosensitizer molecules from complexes. Knowledge of these parameters will make it possible to predict the behavior of albumin and photosensitizer complexes after injection into a medium containing serum.

Authors’ reply:

The determination of the binding constants, as well as the dissociation rate constants of photosensitizer molecules from complexes is quite complicated by the fact that multiple binding may occur for mTHPC. In addition, these constants are extremely dependent to many environmental parameters. For this reason, to answer to the most important issue raised by the reviewer, that is related to the behavior of albumin and photosensitizer complexes after injection into a medium containing serum, we carried out equilibrium dialysis experiments to determine the stability of mTHPC@HSA complex toward a medium containing serum. We also compared the results with the clinical formulation of mTHPC.

We added in the manuscript a sentence that reads:

“The stability of mTHPC@HSA was also measured in a medium containing 10% FBS. Equilibrium dialysis experiments (Figure S8) demonstrated that mTHPC@HSA does not exchange the mTHPC molecule with the free serum proteins, as opposed to the clinical formulation mTHPC which rapidly binds to the serum proteins.”

Comments:

  1. From my point of view, it is of interest to study changes in the properties of solutions of complexes during their storage and dilution.

Authors’ reply:

We carried out the experiments requested by the reviewer and we added in the manuscript a sentence that reads:

“The mTHPC@HSA complex, proved stable for long periods when stored in the dark (at least up to two weeks, Figure S6) and upon dilution (Figure S7).”

Comments:

  1. The use of chemical trap techniques to compare the efficiency of singlet oxygen generation in solutions and in nanostructured materials involves taking into account the distribution of trap molecules in the sample under study. The significant differences in the rate of singlet oxygen generation with the participation of mTHPC and mTHPC-protein established in the article may reflect not so much differences in sensitizing properties as the uneven distribution of trap molecules in the protein solution.

Authors’ reply:

We are well aware about the limitations of chemical trap techniques in the determination of the efficiency of singlet oxygen generation. In our lab we used many techniques to detect singlet oxygen: direct detection of the near-IR luminescence emission of singlet oxygen at 1270 nm, EPR, chemical traps (SOSG, DMA and ABMDMA).

Every technique has pros and cons and, in the end, we think that ABMDMA is the best assay to detect singlet oxygen production because the 9,10-Anthracenediyl-bis(methylene) dimalonic acid (ABMDMA) takes four negative charges in neutral aqueous solutions, and it is highly water soluble. For these reasons it is characterized by even distribution in water.

It does not have the tendency to be trapped inside proteins or micelles/liposomes, as other chemical traps.

The second part of the article is devoted to the study of the processes of sensitization by mTHPC-albumin complexes of cell damage in culture. This part also contains interesting and unusual results. The credibility of these results, as well as the conclusions drawn from them, would be higher if the authors provided information on the following issues.

Comments:

  1. How can mTHPC-albumin complexes increase the delivery of a sensitizer to cells in the presence of a large amount of native albumin in the medium?

Authors’ reply:

The selective uptake of the mTHPC@HSA is due to the fact that in the medium it is used FBS 10%, the “competitive” albumins present in the medium are bovine serum albumins.

Although HSA and BSA yielded complexes comparable in size, they showed different uptake profiles (PLoS ONE 10 (4): e0122581. doi:10.1371/journal.pone.0122581) due to specific recognition systems of HSA that are receptor-mediated.

Human serum albumin confers specificity in the uptake in human cells.

Comments:

  1. From my point of view, in order to understand the features of the interaction of the studied form of the sensitizer with various types of cells, as well as the mechanisms of their photosensitization, it is extremely important to study the quantitative characteristics of the accumulation of mTHPC in cells (for example, by flow cytometry).

Authors’ reply:

According to reviewer suggestion, we included data about the accumulation of mTHPC@HSA and mTHPC/PBS in HNSCC cell lines obtained using flow cytometry. We added necessary information in the “Material and Methods” section (2.3.3 Analysis of intracellular uptake) and in the “Results and Discussion” section (3.4 mTHPC@HSA undergoes intracellular uptake and decreases HNSCC cells viability upon PDT treatment).

Comments:

  1. It is believed that photosensitization of cells by mTHPC is not accompanied by significant damage to the genetic structures of cells. In this regard, what is the reason for the high yield of DNA damage when using albumin and photosensitizer complexes.

Authors’ reply:

We thank the reviewer for his comment. The H2AX phosphorylation represents one of the cellular events involved in the global response to cellular stress, such as the oxidative stress generated after PDT in HNSCC cells. The H2AX phosphorylation is not a test specifically addressed at depicting the genotoxicity and/or mutagenicity of mTHPC@HSA. Our main aim was to investigate the increase in H2AX phosphorylation after phototoxic treatment, as we specify in the revised version of the manuscript. We additionally add proper references and discussion in the paragraph “3.6 mTHPC@HSA induces γ-H2AX upon PDT treatment”.

Reviewer 2 Report

The present manuscript demontstrated that mTHPC@HSA is an alternative PS for HNSCC treatment, with great stability, efficient singlet oxygen  generation  and high dispersion as monomolecular form in HSA. The author have performed elegant methods to chemically characterize the mTHPC@HSA. In addition, the PS was tested in vitro in two HNSCC, showing promising results, with high cytotoxicity, ROS generation and induction of DNA double strand breaks.

The manuscript is well written and easy to follow. 

In the material methods, the authors must include detailed information regarding the irradiation parameters. Only the irradiace is provided. 

In the discussion section, the authors should include a paragraph with the next steps nedded to translate this new formulation of mTHPC in the clinical practice. 

Author Response

Reviewer: 2

The present manuscript demontstrated that mTHPC@HSA is an alternative PS for HNSCC treatment, with great stability, efficient singlet oxygen generation and high dispersion as monomolecular form in HSA. The author have performed elegant methods to chemically characterize the mTHPC@HSA. In addition, the PS was tested in vitro in two HNSCC, showing promising results, with high cytotoxicity, ROS generation and induction of DNA double strand breaks.

The manuscript is well written and easy to follow. 

Comments:

In the material methods, the authors must include detailed information regarding the irradiation parameters. Only the irradiace is provided. 

Authors’ reply:

We followed the suggestion of the reviewer and we added in the “Materials and Methods” section and in the Supporting Information detailed information regarding the irradiation parameters.

We added a sentence in the manuscript that reads:

“The multiwell plate was exposed to the light source (Valex 30W, 6500 K, cold white LED, see figure S1 for the spectral profile of the light source), at 19 cm distance from the cell-plate (irradiance = 24 mW cm−2, energy fluence = 86 J cm-2, measured with the photo-radiometer Delta Ohm LP 471 RAD).”

Comments:

In the discussion section, the authors should include a paragraph with the next steps nedded to translate this new formulation of mTHPC in the clinical practice. 

Authors’ reply:

We thank the reviewer for its useful comment, and we added in the “Conclusion and future perspectives” section the next steps necessary for the translation of the formulation in the clinical practice.

We added a sentence in the manuscript that reads:

“The next steps for the translation from benchside to bedside need the completion of preclinical studies to determine the pharmacokinetics, the toxicological profile, and the PDT performances of the new formulation using in vivo models. Indeed, although our in vitro and in silico studies depict how promising mTHPC@HSA is, further studies are required for its future development.”

Round 2

Reviewer 1 Report

In my opinion, the use of fluorescent characteristics by comparing the emission spectra of complexes and aqueous solutions of mTНPC makes no sense. In aqueous solutions, mTHPC is in an aggregated state and does not fluoresce. At the same time, a comparison of the fluorescence spectra of complexes with solutions of monomeric mTHPC (for example, solutions of mTHFA in ethanol, detergents, etc.) will confirm the complete monomeric sensitizer. I recommend the authors to add the fluorescence spectrum of the equimolar solution of mTGFX in ethanol. 

Comparison of the rates of mTHPC elimination in the course of dialysis for complexes and alcohol solution seems to me not quite correct. If the affinity of mTHPC to the protein is high, then the concentrations of free photosensitizer molecules in the compared samples will be very different. This fact reduces the elimination rate during dialysis, but does not deny the dissociation of complexes and binding of photosensitizer molecules to various structures regardless of the carrier protein.

Despite the fact that I do not agree with some interpretations of the data obtained, I believe that the reviewed article contains new interesting results concerning the distribution mechanisms of photosensitizers and can make a significant contribution to the development of mTHPC based PDT.

Author Response

Reviewer: 1

Comments:

In my opinion, the use of fluorescent characteristics by comparing the emission spectra of complexes and aqueous solutions of mTНPC makes no sense. In aqueous solutions, mTHPC is in an aggregated state and does not fluoresce. At the same time, a comparison of the fluorescence spectra of complexes with solutions of monomeric mTHPC (for example, solutions of mTHFA in ethanol, detergents, etc.) will confirm the complete monomeric sensitizer. I recommend the authors to add the fluorescence spectrum of the equimolar solution of mTGFX in ethanol. 

Authors’ reply:

We followed the suggestion of the reviewer and we compared the UV-Vis/fluorescence spectra of mTHPC@HSA complex with solution of monomeric mTHPC, i.e. mTHPC in the pure clinical formulation (EtOH/PG (40/60, w/w)). When we compared the fluorescence spectra of mTHPC@HSA in PBS and an equimolar solution of mTHPC in EtOH/PG (40/60, w/w), we obtained the same values of fluorescence, demonstrating its monomeric dispersion (Figure 2).

Comments:

Comparison of the rates of mTHPC elimination in the course of dialysis for complexes and alcohol solution seems to me not quite correct. If the affinity of mTHPC to the protein is high, then the concentrations of free photosensitizer molecules in the compared samples will be very different. This fact reduces the elimination rate during dialysis, but does not deny the dissociation of complexes and binding of photosensitizer molecules to various structures regardless of the carrier protein.

Authors’ reply:

We do not deny the possible dissociation of the mTHPC@HSA complex.

Every binding phenomenon is in thermodynamic equilibrium and for sure the mTHPC@HSA complex can dissociate. We demonstrate that the formation of the complex represents a kinetic trap and at least for 24 hours the complex does not exchange mTHPC with serum proteins, maintaining a well-defined biological identity, which is enough for real applications. On the other site the clinical formulation of mTHPC rapidly comes into balance with serum proteins.

Comments

Despite the fact that I do not agree with some interpretations of the data obtained, I believe that the reviewed article contains new interesting results concerning the distribution mechanisms of photosensitizers and can make a significant contribution to the development of mTHPC based PDT.

Authors’ reply:

We thank the reviewer for his/her opinion.